

# Quantifying the association of natal household wealth with women's early marriage in Nepal

Akanksha A. Marphatia[1,4], Naomi M. Saville[2], Dharma S. Manandhar[3], Mario Cortina-Borja[4], Jonathan C. K. Wells[4] and Alice M. Reid[1]

[1] Department of Geography, University of Cambridge, Cambridge, United Kingdom
[2] Institute for Global Health, University College London, London, United Kingdom
[3] Mother and Infant Research Activities, Kathmandu, Nepal
[4] Great Ormond Street Institute of Child Health, University College London, London, United Kingdom

Corresponding authors
Jonathan C. K. Wells,
Jonathan.Wells@ucl.ac.uk
Alice M. Reid, amr1001@cam.ac.uk

## ABSTRACT

**Background:** Women's early marriage (<18 years) is a critical global health issue affecting 650 million women worldwide. It is associated with a range of adverse maternal physical and mental health outcomes, including early childbearing, child undernutrition and morbidity. Poverty is widely asserted to be the key risk factor driving early marriage. However, most studies do not measure wealth in the natal household, but instead, use marital household wealth as a proxy for natal wealth. Further research is required to understand the key drivers of early marriage.

**Methods:** We investigated whether natal household poverty was associated with marrying early, independently of women's lower educational attainment and broader markers of household disadvantage. Data on natal household wealth (material asset score) for 2,432 women aged 18–39 years was used from the cluster-randomized Low Birth Weight South Asia Trial in lowland rural Nepal. Different early marriage definitions (<15, <16, <17 and <18 years) were used because most of our population marries below the conventional 18-year cut-off. Logistic mixed-effects models were fitted to estimate the probabilities, derived from adjusted Odds Ratios, of (a) marrying at different early ages for the full sample and for the uneducated women, and (b) being uneducated in the first place.

**Results:** Women married at median age 15 years (interquartile range 3), and only 18% married ≥18 years. Two-thirds of the women were entirely uneducated. We found that, rather than poverty, women's lower education was the primary factor associated with early marriage, regardless of how 'early' is defined. Neither poverty nor other markers of household disadvantage were associated with early marriage at any age in the uneducated women. However, poverty was associated with women being uneducated.

**Conclusion:** When assets are measured in the natal household in this population, there is no support for the conventional hypothesis that household poverty is associated with daughters' early marriage, but it is associated with not going to school. We propose that improving access to free education would both reduce early

marriage and have broader benefits for maternal and child health and gender
equality.

Natal household poverty, Nepal, South Asia

# INTRODUCTION

Women's early reproduction is detrimental to both maternal and child health (*Finlay,
Özaltin & Canning, 2011*; *Fall et al., 2016*). However, in many societies, strong cultural
norms mean that the vast majority of women marry before having children. In such
societies, early marriage is therefore the gateway to early childbearing. Since early marriage
also has broader implications for women's health, it is a crucial global health issue in its
own right (*Marphatia, Amable & Reid, 2017*).

Globally, 20% of women aged 20–24 years marry or enter into a formal union before
18 years of age (*UNICEF, 2021*). Early marriage is associated with a range of penalties for
women. These include less education, under-nutrition, lower access to contraception
and healthcare, early childbearing, and higher morbidity and mortality during pregnancy
and labor (*Godha, Hotchkiss & Gage, 2013*; *Raj & Boehmer, 2013*; *Ganchimeg et al., 2014*;
*Raj et al., 2014*; *Delprato et al., 2015*; *Goli, Rammohan & Singh, 2015*; *Marphatia et al.,
2021a*; *Wells et al., 2021*). These disadvantages are likely to propagate adverse effects to the
next generation (*Bates, Maselko & Schuler, 2007*; *Marphatia, Amable & Reid, 2017*; *Chari
et al., 2017*).

The United Nations (UN) defines 'child or early marriage' using a cut-off of <18 years
(*UN General Assembly, 2014*, *2018*). However, among the Maithili-speaking Madhesi
population in Nepal, where our study is based, most women marry well below this
threshold, around a median age of 16.5 years (*MOHP, New ERA & ICF International,
2017*). In this population, it would therefore be more informative to investigate the factors
associated with marrying at different early marriage ages.

In most studies using the <18 years cut-off to define early marriage, poverty in women's
natal household, where they are born and raised, is widely suggested to drive early
marriage across the global South, including in Nepal (*ICRW, 2006*; *Chaudhuri, 2015*;
*Hodgkinson, 2016*). However, most studies measure material assets (as an index of wealth)
in the marital household only *after* women have already married, and then use this
information as a proxy for the natal household's wealth *prior* to marriage (*Raj et al., 2014*;
*Delprato et al., 2015*; *Wodon et al., 2017*; *Scott et al., 2021*). This practice therefore
relies on an assumption that women are likely to marry into households of similar wealth
as their natal homes. Wealth of natal and marital households might indeed be correlated,
but a severe shortage of data on natal wealth means that there is little evidence for this
assumption.

The wealth of the natal and marital households might therefore also be uncorrelated.
For example, some studies find that younger marriages (<15 years) may in part be driven

by girls wanting to marry into economically better-off marital households, as a way of escaping the poverty of their natal home (*Human Rights Watch, 2016*). However, as wealth was measured in neither the natal nor marital household in this qualitative study, it is unclear whether the aims of these families were achieved. Moreover, some poorer families are able to educate their daughters to secondary school level, and this is then leveraged to marry them into wealthier households with more educated husbands (*Fafchamps & Shilpi, 2011*; *Jackson, 2012*; *Boyden, 2013*). We argue, therefore, that using wealth of the marital household, where women end up after marriage, to represent wealth of the natal household, where women came from prior to marriage, is inappropriate when investigating the potential role of poverty in driving early marriage.

Attributing wealth to the correct household is crucial in this context, because it reflects not only the family's socio-economic status but also its 'spatial niche'. This term refers to the physical habitat or space within which a household is geographically located, and reflects proximity and accessibility to a range of resources and people (*Marphatia et al., 2021c*). Collectively, these characteristics are likely to shape the natal household's intentions around the timing of their daughter's marriage. Misinterpreting the source of the natal household's interests (by using marital household assets) therefore means that we have an inadequate understanding of how both wealth and other related factors may be associated with early marriage.

Using data on natal households from a cluster-randomized trial in lowland rural Nepal, we investigate the independent associations of natal household poverty, as well as other socio-economic factors, with different age groupings of early marriage. We also investigate the contribution of education to early marriage, and whether broader socio-economic factors are associated with women being uneducated in the first place.

## What do we know about poverty's association with early marriage?

To date, most studies which investigate the link between wealth and age at marriage have focused on women after they have married. Clearly, it is difficult to measure natal household wealth after girls have already married and moved to their marital households. However, these studies assume that marital household assets are a proxy for the natal household's assets (wealth) without appropriate supporting evidence. Studies relating to Nepal vary in whether daughters from the poorest (*Guragain et al., 2017*), or the wealthiest (*Aryal, 2007*) households have the highest risk of marrying early. Other studies find an inconsistent relationship between wealth and marriage age (*Raj et al., 2014*) or no relationship at all (*Pandey, 2017*).

Some studies have used household wealth data appropriately, but in two different ways. One group of studies has described the proportion of women married <18 years stratified by marital wealth quintiles without making an assumption that this also represents wealth in the natal household. These studies consistently find that in impoverished regions of the world, including Nepal, the poorest marital household quintile has the largest proportion of women married <18 years (*ICRW, 2006*; *UNFPA, 2012*; *UNICEF & UNFPA, 2017*; *MacQuarrie & Juan, 2019*). Fewer studies have first measured wealth in the natal household itself, and then investigated if this is associated with the

likelihood of women marrying early (*Muchomba, 2021*). In India, one study found that girls from poorer natal households were more likely to marry early (*Singh & Espinoza Revollo, 2016*), whereas another study, which measured wealth at several time-points from the daughter's birth through to adolescence, found no association between wealth and early marriage (*Marphatia et al., 2021b*). For Nepal, we could only find one such study, which produced a less consistent pattern, whereby it was not the poorest, but the second poorest households whose daughters were married earliest (*Bajracharya & Amin, 2012*). Collectively, these studies suggest that the relationship between poverty and early marriage is not as strong when wealth is measured in natal households, and that using wealth in the marital household as a proxy for wealth in the natal household may be inappropriate.

Many of the insights on how poverty may push girls into early marriage come from qualitative literature, where household wealth is neither measured nor quantified. Nevertheless, such work does offer valuable insights on subjective perspectives and decision-making around marriage. Several studies from Nepal have identified a strong economic rationale for natal households to marry daughters at a young age, related to material poverty: specifically in relation to the costs of caring (food, clothes) for daughters and paying for their education (*Verma, Sinha & Khanna, 2013*; *Chaudhuri, 2015*; *Human Rights Watch, 2016*; *Samuels et al., 2017*). The custom of dowry (illegal, but typically still paid from the natal to marital household) is also likely to contribute to early marriage. Dowry generally demands a substantial proportion of a household's income, and this is greater for families with several daughters (*Sah, 2012*; *Pesando & Abufhele, 2019*). As dowry tends to increase with age and education level, it may lower household investment in daughters schooling and also incentivize early marriage (*Sah, 2012*; *Hodgkinson, 2016*; *Human Rights Watch, 2016*; *Karim, Greene & Picard, 2016*).

Hence, in the poorest families, girls may be perceived as an economic burden and thus the earlier they are married, the better the natal household's economic welfare (*Hodgkinson, 2016*; *Guragain et al., 2017*). However, one study from Nepal reporting both attitudinal and quantitative data found that the primary drivers of early marriage were family pressure, socio-cultural norms, low education and food insecurity; income poverty was cited as a less important factor (*Maharjan et al., 2012*).

## Other markers of disadvantage and early marriage

Beyond low material assets, other markers of disadvantage may also be associated with early marriage. For example, education is a key factor associated with marriage age, and studies generally find that girls' lower educational attainment (years of schooling completed) increases their risk of marrying early (*Raj et al., 2014*; *Delprato et al., 2015*; *Sekine & Hodgkin, 2017*; *Marphatia et al., 2020*; *Scott et al., 2021*). The association between poverty and early marriage may therefore vary by girls' education level, and poverty may also contribute to whether girls are educated in the first place, because of the costs associated with schooling (*e.g.* fees, books, uniforms, *etc.*) (*Verma, Sinha & Khanna, 2013*; *Chaudhuri, 2015*; *Samuels et al., 2017*).

To understand the association between poverty and the timing of women's marriage, we also need to consider other relevant socio-economic factors. In rural contexts, agrarian land-holding is another relevant marker of household wealth (*Fisher & Naidoo, 2016*). Landlessness may increase the risk of food insecurity, which in turn has been associated with both lower schooling and earlier marriage (*Moock & Leslie, 1986*; *UNICEF, 2014*). Caste affiliation is also linked to socio-economic status, with girls from disadvantaged castes generally completing less education and also marrying <18 years (*Stash & Hannum, 2001*; *Sah, 2018*; *Devkota, Eklund & Wagle, 2020*). There is much less literature on the role of the natal household's geographic location in relation to early marriage (*Marphatia et al., 2021c*), but greater distance to school has been found to be a key constraint to accessing education (*Jamison & Lockheed, 1987*; *Ayral, 2014*; *Devkota & Upadhyay, 2015*). Thus, if schooling is not a viable option, marrying daughters early may alleviate household financial pressures and food insecurity (*Maharjan et al., 2012*; *Human Rights Watch, 2016*; *Samuels et al., 2017*).

Socio-cultural norms are also likely to shape both the timing of marriage and the amount of education that girls are likely to complete. *Bicchieri, Jiang & Lindemans (2014)* define these normative social preferences as 'moral rules' that govern decision-making relating to women's life options, whether they refer to marriage, chastity, education, employment, *etc.* Failing to conform to these norms may adversely affect a girl's marital options, and also her natal household's social standing in the community (*Caldwell, Reddy & Caldwell, 1983*; *Maertens, 2011*, *2013*). However, such norms can also change over time. Several studies have suggested that secular changes in attitudes and norms, coupled with widespread advocacy for minimum marriage age legislation, improvements in household wealth, and increased girls' educational attainment, are collectively likely to explain the overall decrease in early marriage over the past ~15 years across South Asia (*Raj, McDougal & Rusch, 2012*; *Allendorf & Thornton, 2015*; *MacQuarrie & Juan, 2019*; *Prakash et al., 2020*; *Scott et al., 2021*).

## Study aim and hypotheses

Our study aims to contribute new insights on the economic and social drivers of women's early marriage and their lack of education in low-income settings. Using objective data on wealth and broader markers of disadvantage measured in the natal household on 2,432 women aged 18–39 years from lowland rural Nepal, we investigate whether natal household poverty is associated with marrying early. We define 'early marriage' using several different age groupings, because most women in our population marry well below the 18-year threshold (universal minimum legal age) conventionally used to define early marriage. This is crucial because if we only use the 18-year threshold, we might miss identifying the factors associated with variability in age at marriage as it is experienced in this early-marrying population.

Since two-thirds of our sample are entirely uneducated, we also investigate whether poverty is associated with early marriage in uneducated women, and whether poverty is associated with women being uneducated in the first place. The uneducated women are interesting to examine on their own because for this group, variability in education cannot

confound the association between wealth and marriage age. Our models adjust for women's age, to capture potential cohort effects and which may indicate secular changes in social norms over time. To ensure observed associations between poverty and early marriage are not an artefact of related factors, we include women's education level as another key exposure, and also broader markers of socio-economic disadvantage measured in women's natal household: agrarian land-holding, geographic location and caste affiliation.

We investigate four hypotheses:

1. that natal household poverty is associated with marrying early, using different ages to define this outcome: <15, <16, <17 and <18 years, in each case compared to marrying ≥18 years;
2. that women's lower educational attainment, independent of natal household poverty and broader markers of socio-economic disadvantage, is associated with early marriage: <15, <16, <17 and <18 years, in each case compared to marrying ≥18 years;
3. that amongst uneducated women, poverty, independent of broader markers of socio-economic disadvantage, is associated with early marriage at different ages;
4. that natal household poverty, independent of broader markers of socio-economic disadvantage, is associated with women being uneducated.

## MATERIALS AND METHODS

Our study is based on data from the Low Birth Weight South Asia Trial (LBWSAT), which assessed the impact of three pregnancy interventions on birth weight and infant growth (*Saville et al., 2018*). This cluster-randomized control trial was conducted across 80 geographic clusters in Dhanusha and Mahottari districts in Province 2 of the Terai region bordering Bihar state in India. Married pregnant women were randomized to one of four intervention arms: Participatory Learning and Action (PLA) behavior change intervention in Women's Groups, PLA with unconditional cash transfers, PLA with a fortified blended food supplement, or a control group accessing Government of Nepal health services. Questionnaires were administered orally to 25,090 married pregnant women aged 10-49 years in the home that they were residing in during pregnancy (*Saville et al., 2016*).

Research ethics approval to conduct the trial was granted by the Nepal Health Research Council (108/2012) and University College London (UCL) Research Ethics Committee (4198/001). Village Development Committee secretaries consented for villages to participate in the trial. Women gave written consent and guardians consented to the participation of married adolescents <18 years of age. Further ethical approvals for secondary analyses of LBWSAT data for this analysis were granted from the Nepal Health Research Council (292/2018), the Research Ethics Committees at UCL (0326/015) and the University of Cambridge (1016).

Marriages in the Maithili-speaking Madhesi population of our study are generally arranged by parents or close relatives, with girls having little say over the timing and choice

of spouse (*Maharjan & Sah, 2012*; *Clarke, 2013*). In 2016, the Maithili-speaking Madhesi women had the lowest median age at marriage (16.5 years) nationwide (*MOHP, New ERA & ICF International, 2017*; *Pandey, 2017*), and were more likely to be uneducated (*Marphatia et al., 2020*). These factors, and gendered socio-cultural norms restricting women's physical mobility outside of the home, mean that women typically have low levels of agency and decision-making power (*Gram et al., 2017*; *Harris-Fry et al., 2018*; *Morrison et al., 2018*). The main livelihood of this population is subsistence farming (rice, wheat, pulses), with the majority of households purchasing some food items from local markets, or 'bazaars' (*Saville, Manandhar & Wells, 2020*).

## Data

### Outcome variables

For our first outcome variable, women's 'early marriage,' we use several different age groupings since the majority of our sample (87%) married below the UN stipulated minimum age of 18 years. In Nepal, the legal minimum age at marriage is 20 years, and until recently, marriage at 18 years was possible with parental permission (*His Majesty's Government of Nepal, 1963*; *Government of Nepal, 2017*). Since our trial was conducted prior to this change in legislation, and as few women had married ≥20 years, we use the minimum marriage age cut-off of 18 years as the reference group. To ensure comparability across these results, the same reference group, marrying ≥18 years, is used irrespective of the age used to define early marriage. We examine the factors associated with marrying <15 years (hence excluding women who married between 15–18 years), <16 years (hence excluding women who married between 16–18 years), <17 years (hence excluding women who married between 17–18 years), and <18 years of age (includes the full sample). In other words, each of the models above the <15 years has the lower group nested within it, but excludes the higher group up to 18 years. Figure 1 illustrates our approach.

Our second outcome variable, women 'being uneducated,' is coded as any formal education (≥1 year) vs no education (0 years). We use this cut-off because two-thirds of our sample have never been to school.

### Exposures

Our primary exposure is the score of assets measured in the women's natal household. The natal household asset score is categorized in quintiles, from 1 (poorest) to 5 (richest). This assessment of assets is widely used by national representative surveys, including in our study context (*MOHP, New ERA & ICF International, 2017*). These assets represent relatively stable markers of wealth, relating to the structure of the home or ownership of consumer goods that require significant financial outlay. They are assumed already to exist before a daughter marries. However, in contrast to the 12-asset score that has been used in previous studies on this population (*Saville et al., 2016*; *MOHP, New ERA & ICF International, 2017*; *Sah, 2018*), we use an 8-asset score. We exclude goods such as color television, motorbike, and computer because they could have been acquired after the daughter had married, especially if there was a long time-gap between marriage and

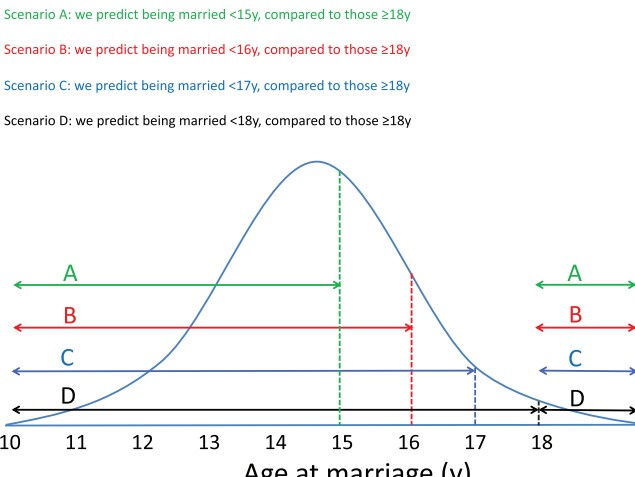

**Figure 1** **Early marriage groups used in analysis.** Our first outcome variable, women's 'early marriage,' uses four different age groupings, described as Scenarios A, B, C and D. To ensure comparability across these results, the same reference group, marrying at the minimum age cut-off of ≥18 years, is used irrespective of the age used to define early marriage. Each of the scenarios above the <15 years has the lower group nested within it, but excludes the higher group up to 18 years.

when the assets were measured. We also exclude agrarian land-holding from the score, because we want to investigate whether its association with women's early marriage and lack of education is independent to that of assets.

We therefore construct our score of assets from eight variables using principal component analysis (*Vyas & Kumaranayake, 2006*). The first principal component has positive factor loadings for all eight variables and accounts for 36.7% of the variability, compared to 13.2% for the second principal component. Thus we use the first principal component as the marker of natal household wealth. The eight variables contributing the highest factor loadings to the first principal component, listed in order of decreasing size (weight shown in parenthesis), are: wall (0.457), roofing (0.442) and flooring (0.423) materials, toilet facilities (0.412), number of rooms used for sleeping in the house (0.317), access to electricity (0.242), drinking water source (0.227) and non-biomass cooking fuel use (0.188).

Our second key exposure is women's educational attainment. Education is coded according to the Nepalese education system: none; primary (1–5 years); lower-secondary (6–8 years); or secondary or higher (≥9 years) (*Ministry of Education Nepal, 2016*).

We also include broader markers of household socio-economic disadvantage. Agrarian land-holding is coded as none, 0.01 to 0.5, 0.51 to 0.99 and ≥1 hectare. A household's spatial niche, defined by its geographic location, is categorized by accessibility to large markets, known as 'bazaars,' using the normal form of transport, quantified in terms of time (<30 min; 30–59 min; 60–89 min; or ≥90 min). Large bazaars may be a proxy for access to broader social connections, resources, larger health facilities and schools. Four groups describe caste affiliation: disadvantaged castes are coded into two separate groups: Dalit and Muslim, middle combines Janjati and various Madhesi castes, and

advantaged combines Yadav and Brahmin. We do not include religion as the disadvantaged caste variable already examines the Muslim faith separately, and all other groups refer to the Hindu faith.

The women in our sample range from age 18 to age 39 years. Access to education and social norms may have changed in the 20 years between the oldest and youngest generations. We therefore control for women's age to capture the ways in which secular changes in societal norms may impact marriage age and education patterns.

## Statistical methods

We first test for bias in the characteristics of women with assets measured in their natal *versus* marital households using chi-squared tests (categorical variables) and non-parametric *k*-sample analysis of variance (Kruskal–Wallis test; continuous variables). We also test for differences in individual assets by asset quintile level using chi-squared tests (categorical variables) and ANOVA (reporting the mean and standard deviation, SD). Given the skewed distribution of women's age, we report the median (and interquartile range, IQR) values in completed integer years. A heat table examines the distribution of women by education level and natal household wealth quintiles. We use SPSS 26 to conduct these analyses (IBM Corp., Armonk, NY, USA). Using R library tidyverse and ggplot2, we create boxplots to stratify the association of women's marriage age with their education by natal household asset quintiles (*Wickham, 2016*; *Wickham et al., 2019*).

We fit logistic mixed-effects models with a random effect on the intercept accounting for within-cluster variability. If $i$ denotes individuals nested within geographic clusters, the equation for a general regression model of this form with $p$ covariates, and random effects in the intercept ($u_{i0}$) and in one covariate ($u_{i1}$) is shown below. Logit is the log-odds function, defined as $\ln\left(\frac{\pi_i}{1-\pi_i}\right)$, where $\pi_i$ is the conditional probability that the binary outcome variable $Y_i$ equals one divided by the probability that it equals zero. $\beta_0$ is the intercept (constant term) and $\beta_1$ is the slope, or coefficient estimate describing the relationship between the outcome and the predictor variable $X_1$. The index $j$ denotes variables, respectively, and the residual error term, $\varepsilon_i$ is assumed to be Normally distributed with mean 0 and variance $\sigma^2$, whilst the random effects are assumed to follow a bivariate Normal distribution with mean (0,0) and variance-covariance matrix $\begin{pmatrix} \vartheta_0^2 & \tau_{01} \\ \tau_{01} & \vartheta_1^2 \end{pmatrix}$, independently of $\varepsilon$.

Our logistic mixed-effects model, shown below differs from the usual fixed-effects model in that it includes random effects accounting for unobserved heterogeneity due to the 80 geographic clusters of the trial:

$$\text{logit} = \ln(odds) = \ln\left(\frac{\pi_i}{1 - \pi_i}\right) = (\beta_0 + u_{i0}) + (\beta_1 + u_{i1})x_{i1} + \sum_{j=2}^{p} \beta_j\, x_{ij} + \varepsilon_i$$

Our models estimate the probabilities, derived from adjusted Odds Ratios (aORs) with 95% Confidence Interval (CI) of women (a) marrying <15, <16 <17 or <18 years, and (b) being uneducated. Models of the factors associated with being uneducated do not
include marriage age because it is not appropriate to use a factor which occurred at time B (marriage age) to predict something earlier at time A (never starting school). All of the models control for women's age (potential cohort effect), which, when included together with their age at marriage effectively accounts for the time-gap between when they married and their current age (when they were recruited into the trial). Since our interest is in understanding whether poverty (defined relative to the richest quintile) is associated with early marriage, the reference group is set as the highest category across variables, hence: richest asset quintile, secondary education, agrarian land-holding of ≥1 hectare and advantaged caste. Living near to the biggest bazaar (<30 min) is set as the reference group because we assume it is a proxy for better access to school and other resources.

We evaluate goodness-of-fit using the Nakagawa–Schielzeth marginal $R^2$ which measures the percentage of variance explained by the model's fixed effects (*Nakagawa & Schielzeth, 2013*). Logistic mixed-effects models are fitted using the R library `lme4` (*Bates et al., 2014*).

We adjust for trial arm because women from natal households were more likely to enroll in the cash and food interventions. However, since the trial recruited women who were already married and currently pregnant, the intervention could not have influenced marriage age or education (which typically is ended before/once women marry). Moreover, assets were measured before the trial was conducted, so the cash supplementation arm could not have changed their value. As the trial arm was not associated with our outcomes, we do not report the findings, although it is still controlled for in our analyses.

We conduct a sensitivity analysis to examine if the association between poverty and our outcomes change if we apply looser selection criteria, by including all of the women measured in their natal household ($n = 3,379$) regardless of their age. While this introduces the possibility of distortion due to selection bias (as discussed above), it nevertheless allows us to test our hypotheses with a much bigger sample of women.

## RESULTS

### Sample selection

Married pregnant women were interviewed in the home in which they were residing at the time of recruitment into the trial. Of the 25,090 women recruited into our study, we first exclude 408 women with multiple pregnancies during the trial to ensure they are not double counted in our analysis. Second, of the remaining 24,682 women (raw data in Data S1 contains this sample), we exclude 3,968 women who had no data on the household in which assets were measured and a further 17,335 women who were interviewed in their marital home. This leaves 3,379 women whose assets and other characteristics were measured in their natal household.

Third, we exclude 947 women aged <18 years because they would not have had the chance of marrying at older ages, nor adequate time to finish greater levels of education before marrying. The relationship between wealth, education and marriage age is not

likely to be distorted due to the selection bias of recruiting only married pregnant women into our study. In the context of our study, women are very unlikely to continue their education after marriage (*Sekine & Hodgkin, 2017*). The vast majority of married women are also likely to have children, and it is common in Nepal get pregnant in the first 2 years after marriage (*MacQuarrie, 2016*; *Marphatia et al., 2020*). Our analysis therefore includes 2,432 women aged 18-39 years measured in their natal households, representing 9.7% of the total 25,090 women recruited into the trial.

Table S1 examines whether the characteristics of the women measured in their natal homes are different to those measured in their marital homes. This enables us to identify potential bias in our sample which may result in these two groups of women having a different relationship between wealth and marriage age. A direct comparison between these women is not possible because we do not know the socio-economic background of the natal households of the women who were measured in their marital home.

In comparison to women who were measured in their marital home, those measured in their natal homes are younger, have been married for less time, but at an older age. A greater proportion of women measured in their natal households are from disadvantaged castes, residing further away from big bazaars, and in the trial's cash and food supplementation arms. It is possible that the trial may have incentivised women to return to their natal homes to access these incentives. Women did not differ in their education level, age at first pregnancy, or household asset score. Whilst these results show that women who went to their natal homes during pregnancy are different than those who stayed in their marital homes, the differences are small. The women in our sample still represent an important sub-group of women with rare data on the socio-economic characteristics of their natal homes.

## Description of sample

Table 1 describes our sample of women aged 18–39 years. Women are of median age 21 years (IQR 4) and have been married for a median of 5 years (IQR 5). Marriage is typically early among women (median 15 years, IQR 3), and 18% married ≥18 years. Two-thirds of the women are uneducated, 39% are from natal households without agrarian land and disadvantaged castes respectively, and 29% live far from large bazaars.

The heat table shows the overall number of women by their education level and natal household wealth quintiles (Fig. 2). Green shaded areas indicate low numbers and red shaded numbers the highest numbers. Within the whole sample, uneducated women are most likely to come from poorer natal households, and the more educated women are more likely to come from richer households. However, since two-thirds of our sample is uneducated, there is substantial variability by wealth that is independent of education.

Figure 3 illustrates the association of women's marriage age with their education level, stratified by natal household asset quintiles. Overall, within each wealth group, women with more education marry later. The level of education associated with delayed marriage differs slightly across wealth groups. Interestingly, even in the poorest and poor wealth groups, the median age at marriage is 2 years later among women with ≥6 years of education than among those with no or less education.

**Table 1 Description of sample.**

| | Traits measured in women's natal household ( $n = 2{,}432$) | |
| --- | --- | --- |
| | Median | IQR |
| Women's age (y) | 21 | 4 |
| Women's age at marriage (y) | 15 | 3 |
| Time since marriage (y) | 5 | 5 |
| | **Frequency** | **%** |
| Trial intervention arm | | |
| Control | 429 | 17.6 |
| Women's Group | 446 | 18.3 |
| Women's Group with cash transfer | 793 | 32.6 |
| Women's Group with food supplement | 764 | 31.4 |
| Women's age at marriage (y) | | |
| <15 years | 666 | 27.4 |
| 15 years | 574 | 23.6 |
| 16 years | 356 | 14.6 |
| 17 years | 408 | 16.8 |
| ≥18 years | 428 | 17.6 |
| Women's education level (y) | | |
| None | 1,610 | 66.2 |
| Primary (1–5 years) | 244 | 10.0 |
| Lower-secondary (6–8 years) | 196 | 8.1 |
| Secondary or higher (≥9 years) | 382 | 15.7 |
| Natal household asset score (quintiles) | | |
| 1: poorest | 525 | 21.6 |
| 2: 2nd poorest | 484 | 19.9 |
| 3: middle | 528 | 21.7 |
| 4: 2nd richest | 465 | 19.1 |
| 5: richest | 430 | 17.7 |
| Natal household agrarian land-holding | | |
| None | 951 | 39.1 |
| 0.01 to 0.5 hectares | 750 | 30.8 |
| 0.51 to 0.99 hectares | 344 | 14.1 |
| ≥1 hectare | 387 | 15.9 |
| Natal household access to big bazaar | | |
| <30 min | 798 | 32.8 |
| 30–59 min | 934 | 38.4 |
| 60–89 min | 472 | 19.4 |
| ≥90 min | 228 | 9.4 |
| Natal household caste affiliation | | |
| Disadvantaged: Dalit | 494 | 20.3 |
| Disadvantaged: Muslim | 468 | 19.2 |
| Middle: Janjati, Terai castes | 927 | 38.1 |
| Advantaged: Yadav, Brahmin | 543 | 22.3 |

**Note:**
IQR, interquartile range.

| Women's education (y) | Natal household asset quintiles | | | | | |
|---|---|---|---|---|---|---|
| | Poorest | 2nd Poorest | Mid-level | 2nd Richest | Richest | Total row |
| None | 464 | 351 | 356 | 255 | 184 | 1,610 |
| Primary (1-5 years) | 27 | 50 | 61 | 56 | 50 | 244 |
| Lower-secondary (6-8 years) | 20 | 38 | 41 | 59 | 38 | 196 |
| Secondary/higher (≥9 years) | 14 | 45 | 70 | 95 | 158 | 382 |
| Total column | 525 | 484 | 528 | 465 | 430 | 2,432 |

**Figure 2 Heat map of women's educational attainment by natal household wealth.** Green shaded areas indicate low numbers and red shaded numbers the highest numbers.

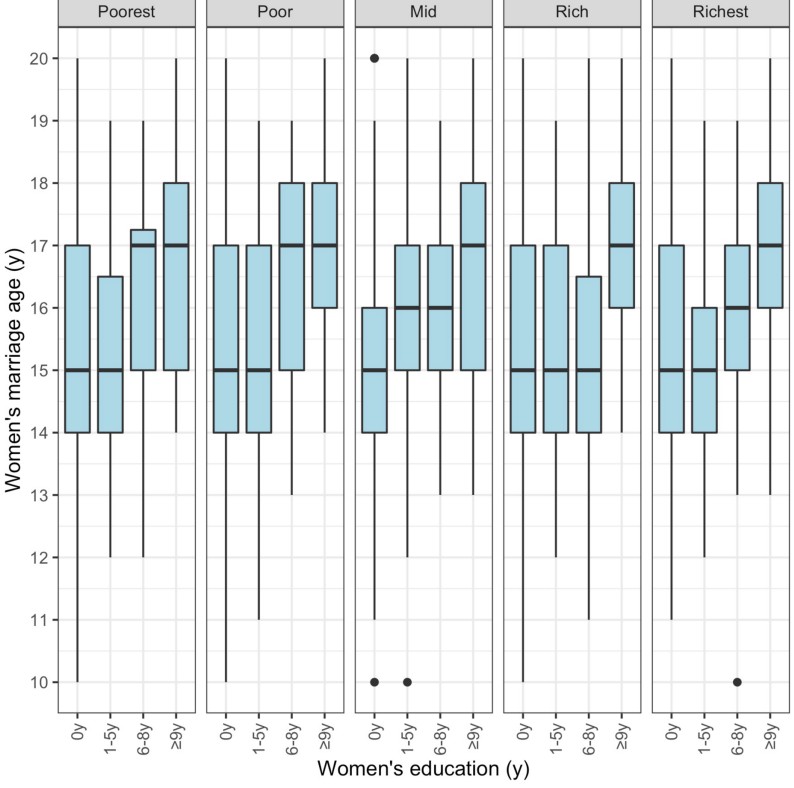

**Figure 3 Association of women's marriage age and their education level stratified by natal household asset score.** This figure uses the raw data to stratify the association of women's marriage age with women's education by natal household asset quintiles.

## Hypothesis 1

Hypothesis 1 investigates the association of the natal household asset score with the odds of early marriage, using each of the four age-definitions of early marriage. Across Models 1–4 in Table 2, there is no evidence of a cohort effect, which may be explained by the relatively narrow age range of our sample. Relative to the richest asset quintile, women from all other asset quintiles have an elevated risk of early marriage, regardless of the age threshold used to define 'early'. However, there is little consistent gradient between the coefficients of the poorest four asset groups, indicating that the main difference in marriage age is between the richest and the rest (this is confirmed by analyses using other asset groups as reference (not shown)). The variance in marriage age explained by the four

**Table 2 Hypothesis 1: Associations of natal household asset score with marrying early.**

| | Model 1: Marrying <15 years $n = 1,094^1$ $R^2 = 0.050$ | | Model 2: Marrying <16 years $n = 1,668^2$ $R^2 = 0.032$ | | Model 3: Marrying <17 years $n = 2,024^3$ $R^2 = 0.028$ | | Model 4: Marrying <18 years $n = 2,432^4$ $R^2 = 0.021$ | |
|---|---|---|---|---|---|---|---|---|
| | aOR (95% CI) | p-value | aOR (95% CI) | p-value | aOR (95% CI) | p-value | aOR (95% CI) | p-value |
| Women's age (y) | 1.02 [0.97–1.06] | 0.485 | 1.01 [0.97–1.05] | 0.556 | 0.99 [0.96–1.03] | 0.646 | 0.98 [0.95–1.01] | 0.178 |
| Asset score | | | | | | | | |
| Poorest | 2.98 [1.90–4.67] | <0.001 | 2.44 [1.67–3.58] | <0.001 | 2.26 [1.58–3.24] | <0.001 | 1.93 [1.37–2.72] | <0.001 |
| 2nd poorest | 2.39 [1.53–3.71] | <0.001 | 1.68 [1.16–2.44] | 0.007 | 1.57 [1.11–2.23] | 0.002 | 1.41 [1.01–1.97] | 0.043 |
| Mid | 2.12 [1.35–3.31] | 0.001 | 1.83 [1.26–2.66] | 0.001 | 1.74 [1.23–2.47] | 0.001 | 1.52 [1.09–2.12] | 0.013 |
| 2nd richest | 2.34 [1.49–3.69] | <0.001 | 1.93 [1.31–2.83] | 0.001 | 1.90 [1.32–2.73] | <0.001 | 1.72 [1.22–2.43] | 0.002 |
| Richest (ref) | 1.00 | | 1.00 | | 1.00 | | 1.00 | |
| Intercept | 0.52 [0.16–1.72] | 0.286 | 1.46 [0.53–3.97] | 0.462 | 3.11 [1.23–7.86] | 0.017 | 6.00 [2.54, 14.15] | <0.001 |

**Notes:**

Models include fixed and random effects estimates for geographic clusters and control for trial arm. Grey shading indicates statistically significant associations. As associations of trial arm with early marriage across the age groupings were not statistically significant, they are not reported in Tables. aOR, adjusted Odds Ratio. CI, 95% Confidence Interval.

[1] $n = 428$ married ≥18 years vs $n = 666$ married <15 years.
[2] $n = 428$ married ≥18 years vs $n = 1,240$ married <16 years.
[3] $n = 428$ married ≥18 years vs $n = 1,596$ married <17 years.
[4] $n = 428$ married ≥18 years vs $n = 2,004$ married <18 years.

models is very low: 5.0%, 3.2%, 2.8% and 2.1%. These results do not support our first hypothesis, that natal household poverty is associated with the likelihood of marrying at different early ages.

## Hypothesis 2

Hypothesis 2 investigates the association of broader socio-economic factors with the likelihood of marrying early as defined by the four different age thresholds. With the exception of Models 3 and 4, age is not a significant factor associated with early marriage, indicating the absence of a cohort effect (Table 3). Models 3 and 4 shows that older women are less likely to marry <18 years than ≥18 years, which is perhaps surprising: we expected that further in the past (older women) prevailing socio-cultural norms might have more strongly favoured earlier marriage. This cohort effect only emerges when education is controlled for in the analysis, which suggests that it is connected to difference in the availability of, and attitudes towards, education over time. It is also possible that this cohort effect emerges because in the past, high levels of education were rarer, and therefore more tightly associated with delayed marriage compared to more recent times, when education is more widely available to the younger women in our sample.

Across Models 1–4 in Table 3, the asset score is no longer associated with early marriage when women's education is included. The effect of lower wealth appears therefore to be channelled through women's education, which is a better predictor of early marriage in this population. There is a clear education gradient in the likelihood of marrying early, however 'early' is defined. Relative to secondary education, all other education levels have an elevated risk of early marriage, with uneducated women having a substantially greater risk. The magnitude of effect of education also decreases with each additional year used to define early marriage, suggesting that education tends to increase with age, but

**Table 3 Hypothesis 2: Broader socio-economic factors associated with women marrying early.**

| | Model 1: Marrying <15 years $n = 1{,}094$[1] $R^2 = 0.235$ | | Model 2: Marrying <16 years $n = 1{,}668$[2] $R^2 = 0.164$ | | Model 3: Marrying <17 years $n = 2{,}024$[3] $R^2 = 0.131$ | | Model 4: Marrying <18 years $n = 2{,}432$[4] $R^2 = 0.106$ | |
|---|---|---|---|---|---|---|---|---|
| | aOR (95% CI) | p-value | aOR (95% CI) | p-value | aOR (95% CI) | p-value | aOR (95% CI) | p-value |
| Women's age (y) | 0.99 [0.94–1.04] | 0.670 | 0.98 [0.94–1.02] | 0.401 | 0.96 [0.93–1.00] | 0.047 | 0.95 [0.91–0.98] | 0.003 |
| Asset score | | | | | | | | |
| Poorest | 1.04 [0.58–1.85] | 0.902 | 0.90 [0.55–1.44] | 0.651 | 0.98 [0.63–1.53] | 0.933 | 0.92 [0.60–1.39] | 0.675 |
| 2nd poorest | 1.02 [0.59–1.74] | 0.949 | 0.74 [0.47–1.15] | 0.184 | 0.82 [0.55–1.23] | 0.338 | 0.79 [0.54–1.16] | 0.233 |
| Mid | 0.95 [0.56–1.63] | 0.863 | 0.91 [0.59–1.40] | 0.658 | 1.01 [0.68–1.50] | 0.948 | 0.94 [0.65–1.37] | 0.763 |
| 2nd richest | 1.61 [0.94–2.75] | 0.083 | 1.30 [0.84–2.02] | 0.233 | 1.48 [1.00–2.21] | 0.051 | 1.31 [0.90–1.89] | 0.157 |
| Richest (ref) | 1.00 | | 1.00 | | 1.00 | | 1.00 | |
| Women's education | | | | | | | | |
| None | 18.43 [10.60–32.03] | <0.001 | 11.50 [7.66–17.28] | <0.001 | 8.00 [5.59–11.47] | <0.001 | 5.65 [4.07–7.84] | <0.001 |
| Primary (1–5 years) | 11.72 [6.06–22.68] | <0.001 | 7.21 [4.30–12.08] | <0.001 | 5.07 [3.18–8.08] | <0.001 | 3.81 [2.46–5.89] | <0.001 |
| Lower-secondary (6–8 years) | 5.23 [2.56–10.66] | <0.001 | 4.56 [2.64–7.88] | <0.001 | 3.96 [2.43–6.45] | <0.001 | 3.43 [2.18–5.41] | <0.001 |
| Secondary/higher (≥9) (ref) | 1.00 | | 1.00 | | 1.00 | | 1.00 | |
| Agrarian land | | | | | | | | |
| None | 1.05 [0.61–1.79] | 0.866 | 1.06 [0.68–1.65] | 0.793 | 0.95 [0.63–1.43] | 0.819 | 0.98 [0.67–1.44] | 0.927 |
| 0.01 to 0.5 hectares | 1.36 [0.81–2.26] | 0.242 | 1.42 [0.93–2.15] | 0.104 | 1.26 [0.85–1.85] | 0.246 | 1.32 [0.92–1.89] | 0.136 |
| 0.51 to 0.99 hectares | 0.96 [0.55–1.69] | 0.895 | 1.08 [0.68–1.70] | 0.746 | 1.09 [0.72–1.65] | 0.688 | 1.09 [0.74–1.60] | 0.671 |
| ≥1 hectare (ref) | 1.00 | | 1.00 | | 1.00 | | 1.00 | |
| Access to big bazaar | | | | | | | | |
| <30 min (ref) | 1.00 | | 1.00 | | 1.00 | | 1.00 | |
| 30–59 min | 1.09 [0.71–1.68] | 0.682 | 1.15 [0.80–1.64] | 0.443 | 1.03 [0.74–1.44] | 0.841 | 1.09 [0.80–1.48] | 0.605 |
| 60–89 min | 1.11 [0.66–1.87] | 0.702 | 1.17 [0.76–1.80] | 0.471 | 1.01 [0.68–1.51] | 0.954 | 1.01 [0.70–1.46] | 0.967 |
| ≥90 min | 1.32 [0.69–2.53] | 0.409 | 1.14 [0.65–2.01] | 0.650 | 0.90 [0.53–1.52] | 0.687 | 1.03 [0.63–1.68] | 0.912 |
| Caste | | | | | | | | |
| Disadvantaged: Dalit | 0.98 [0.56–1.71] | 0.931 | 1.13 [0.71–1.78] | 0.616 | 1.19 [0.78–1.83] | 0.425 | 1.20 [0.80–1.79] | 0.376 |
| Disadvantaged: Muslim | 1.01 [0.59–1.71] | 0.984 | 0.99 [0.63–1.55] | 0.956 | 1.01 [0.67–1.55] | 0.946 | 1.03 [0.69–1.54] | 0.883 |
| Middle: Janjati, Terai castes | 0.99 [0.63–1.54] | 0.953 | 1.13 [0.78–1.62] | 0.517 | 1.13 [0.81–1.58] | 0.467 | 1.09 [0.80–1.49] | 0.576 |
| Advantaged: Yadav, Brahmin (ref) | 1.00 | | 1.00 | | 1.00 | | 1.00 | |
| Intercept | 0.14 [0.03–0.59] | 0.007 | 0.55 [0.17–1.77] | 0.316 | 1.63 [0.57–4.72] | 0.363 | 4.01 [1.53–10.52] | 0.005 |

**Notes:**
Models include fixed and random effects estimates for geographic clusters and control for trial arm. Grey shading indicates statistically significant associations.
aOR, adjusted Odds Ratio. CI, 95% Confidence Interval.
[1] $n = 428$ married ≥18 years *vs* $n = 666$ married <15 years.
[2] $n = 428$ married ≥18 years *vs* $n = 1{,}240$ married <16 years.
[3] $n = 428$ married ≥18 years *vs* $n = 1{,}596$ married <17 years.
[4] $n = 428$ married ≥18 years *vs* $n = 2{,}004$ married <18 years.

assets do not. Neither land, caste nor geographic location are associated with any of the marriage age models when women's education is controlled.

Compared to Table 2, these models are almost three times better at explaining the variance across the four marriage age groups: 23.5%, 16.4%, 13.1% and 10.6%.

**Table 4 Hypothesis 3: Associations of natal household asset score and marrying early in uneducated women.**

| | Model 1: Marrying <15 years $n = 736$[1] $R^2 = 0.062$ | | Model 2: Marrying <16 years $n = 1,156$[2] $R^2 = 0.054$ | | Model 3: Marrying <17 years $n = 1,376$[3] $R^2 = 0.050$ | | Model 4: Marrying <18 years $n = 1,610$[4] $R^2 = 0.050$ | |
|---|---|---|---|---|---|---|---|---|
| | aOR (95% CI) | p-value | aOR (95% CI) | p-value | aOR (95% CI) | p-value | aOR (95% CI) | p-value |
| Women's age (y) | 1.00 [0.94–1.06] | 0.951 | 1.00 [0.95–1.05] | 0.923 | 0.98 [0.94–1.03] | 0.474 | 0.98 [0.93–1.02] | 0.316 |
| Asset score | | | | | | | | |
| Poorest | 0.89 [0.46–1.72] | 0.733 | 0.87 [0.48–1.57] | 0.634 | 0.91 [0.52–1.61] | 0.756 | 0.82 [0.48–1.41] | 0.470 |
| 2nd poorest | 0.83 [0.43–1.62] | 0.590 | 0.69 (0.37–1.27] | 0.231 | 0.70 [0.39–1.25] | 0.228 | 0.67 [0.38–1.16] | 0.155 |
| Mid | 0.87 [0.44–1.71] | 0.679 | 0.83 [0.45–1.54] | 0.560 | 0.83 [0.46–1.50] | 0.544 | 0.75 [0.43–1.32] | 0.319 |
| 2nd richest | 1.71 [0.80–3.65] | 0.168 | 1.53 [0.76–3.11] | 0.235 | 1.48 [0.75–2.93] | 0.258 | 1.48 [0.77–2.84] | 0.241 |
| Richest (ref) | 1.00 | | 1.00 | | 1.00 | | 1.00 | |
| Intercept | 3.28 [0.70–15.41] | 0.133 | 7.14 [1.74–29.26] | 0.006 | 12.28 [3.27–46.17] | <0.001 | 17.30 [4.98–60.07] | <0.001 |

**Notes:**

Models include fixed and random effects estimates for geographic clusters and control for trial arm. aOR, adjusted Odds Ratio. CI, 95% Confidence Interval.

[1] $n = 206$ married ≥18 years *vs* $n = 530$ married <15 years.
[2] $n = 206$ married ≥18 years *vs* $n = 950$ married <16 years.
[3] $n = 206$ married ≥18 years *vs* $n = 1,170$ married <17 years.
[4] $n = 206$ married ≥18 years *vs* $n = 1,404$ married <18 years.

These results support our second hypothesis, that women's lower educational attainment, independent of natal household poverty and broader markers of socio-economic disadvantage, is associated with early marriage, whatever the age threshold used to define 'early'.

## Hypothesis 3

Hypothesis 3 investigates whether amongst the uneducated women ($n = 1,610$), poverty, independent of broader markers of socio-economic disadvantage, is associated with early marriage at different ages. Results show no cohort effect, nor any association with wealth (Table 4 Models 1–4), suggesting that the uneducated women who marry early are not systematically more or less wealthy compared to those marrying ≥18 years. There is also no association between broader socio-economic factors and early marriage, whatever age threshold is used to define early marriage (Table 5 Models 1–4).

In Table 4, models explain 6.2%, 5.4%, 5.0% and 5.0% of the variance in women marrying <15, <16, <17 years and <18 years respectively. In Table 5, models explain slightly more of the variance in early marriage across the age groups 7.1%, 6.6%, 5.8% and 6.0%. In comparison to models in Table 3 for women with all educational levels, those in Table 5 explain less variance. The total absence of women's education in these models is likely to explain this difference. These results do not support our third hypothesis, that among uneducated women, poverty, independent of broader markers of socio-economic disadvantage, is associated with early marriage at different ages.

## Hypothesis 4

Hypothesis 4 investigates whether poverty, independent of broader socio-economic factors, is associated with women being uneducated. Across the two models in Table 6, older women are more likely to be uneducated, reflecting the increase in the availability

**Table 5 Hypothesis 3: Broader socio-economic factors associated with women marrying early in uneducated women.**

| | Model 1: Marrying <15 years n = 736[1] $R^2$ = 0.071 | | Model 2: Marrying <16 years n = 1,156[2] $R^2$ = 0.066 | | Model 3: Marrying <17 years n = 1,376[3] $R^2$ = 0.058 | | Model 4: Marrying <18 years n = 1,610[4] $R^2$ = 0.060 | |
|---|---|---|---|---|---|---|---|---|
| | aOR (95% CI) | *p*-value | aOR (95% CI) | *p*-value | aOR (95% CI) | *p*-value | aOR (95% CI) | *p*-value |
| Women's age (y) | 1.00 [0.95–1.06] | 0.959 | 1.00 [0.95–1.05] | 0.995 | 0.98 [0.94–1.03] | 0.482 | 0.98 [0.93–1.02] | 0.338 |
| Asset score | | | | | | | | |
| Poorest | 0.75 [0.36–1.57] | 0.449 | 0.78 [0.40–1.49] | 0.446 | 0.86 [0.46–1.61] | 0.639 | 0.78 [0.43–1.40] | 0.400 |
| 2nd poorest | 0.74 [0.37–1.50] | 0.402 | 0.63 [0.33–1.20] | 0.163 | 0.65 [0.35–1.20] | 0.172 | 0.62 [0.35–1.12] | 0.115 |
| Mid | 0.79 [0.39–1.59] | 0.503 | 0.77 [0.41–1.46] | 0.432 | 0.79 [0.43–1.45] | 0.439 | 0.72 [0.40–1.28] | 0.260 |
| 2nd richest | 1.56 [0.72–3.38] | 0.262 | 1.45 [0.71–2.96] | 0.311 | 1.41 [0.71–2.82] | 0.325 | 1.43 [0.74–2.77] | 0.287 |
| Richest (ref) | 1.00 | | 1.00 | | 1.00 | | 1.00 | |
| Agrarian land | | | | | | | | |
| None | 1.41 [0.72–2.78] | 0.315 | 1.24 [0.69–2.25] | 0.476 | 1.16 [0.65–2.06] | 0.609 | 1.26 [0.73–2.18] | 0.402 |
| 0.01 to 0.5 hectares | 1.59 [0.81–3.11] | 0.179 | 1.48 [0.82–2.68] | 0.196 | 1.46 [0.82–2.59] | 0.200 | 1.67 [0.97–2.89] | 0.066 |
| 0.51 to 0.99 hectares | 1.21 [0.55–2.67] | 0.634 | 1.18 [0.58–2.39] | 0.646 | 1.18 [0.60–2.31] | 0.638 | 1.32 [0.69–2.53] | 0.395 |
| ≥1 hectare (ref) | 1.00 | | 1.00 | | 1.00 | | 1.00 | |
| Access to big bazaar | | | | | | | | |
| <30 min (ref) | 1.00 | | 1.00 | | 1.00 | | 1.00 | |
| 30–59 min | 0.94 [0.56–1.57] | 0.810 | 0.91 [0.57–1.44] | 0.687 | 0.89 [0.57–1.38] | 0.597 | 0.92 [0.61–1.41] | 0.715 |
| 60–89 min | 1.10 [0.58–2.09] | 0.777 | 1.35 [0.76–2.43] | 0.308 | 1.19 [0.68–2.07] | 0.536 | 1.18 [0.70–2.00] | 0.536 |
| ≥90 min | 1.47 [0.67–3.22] | 0.331 | 1.29 [0.61–2.70] | 0.506 | 1.09 [0.54–2.20] | 0.803 | 1.10 [0.56–2.14] | 0.790 |
| Caste | | | | | | | | |
| Disadvantaged: Dalit | 1.09 [0.54–2.20] | 0.800 | 1.03 [0.56–1.89] | 0.933 | 1.01 [0.56–1.82] | 0.973 | 1.07 [0.61–1.88] | 0.813 |
| Disadvantaged: Muslim | 0.96 [0.50–1.82] | 0.890 | 0.85 [0.48–1.51] | 0.576 | 0.87 [0.50–1.50] | 0.610 | 0.88 [0.52–1.49] | 0.642 |
| Middle: Janjati, Terai castes | 0.96 [0.53–1.74] | 0.904 | 1.00 [0.59–1.69] | 0.996 | 1.00 [0.60–1.66] | 0.996 | 0.96 [0.59–1.56] | 0.865 |
| Advantaged: Yadav, Brahmin (ref) | 1.00 | | 1.00 | | 1.00 | | 1.00 | |
| Intercept | 2.27 [0.40–12.91] | 0.354 | 5.62 [1.16–27.28] | 0.032 | 10.94 [2.48–48.26] | 0.002 | 13.49 [3.31–55.02] | <0.001 |

Notes:
Models include fixed and random effects estimates for geographic clusters and control for trial arm. aOR, Adjusted Odds Ratio. CI, 95% Confidence Interval.
[1] n = 206 married ≥18 years *vs* n = 530 married <15 years.
[2] n = 206 married ≥18 years *vs* n = 950 married <16 years.
[3] n = 206 married ≥18 years *vs* n = 1,170 married <17 years.
[4] n = 206 married ≥18 years *vs* n = 1,404 married <18 years.

and acceptability of education for girls over the 20-year period during which our sample was maturing.

Model 1 shows a clear wealth gradient in the likelihood of being uneducated. Relative to the richest quintile, all other asset quintiles have an elevated risk of being uneducated, with the poorest quintile having a substantially greater risk. Model 2 finds a similar pattern after inclusion of broader socio-economic factors. In comparison to Model 1, the magnitude of effect of the first wealth quintile is weaker, that of the mid quintile only slightly weaker, and unchanged in the two richest quintiles. Relative to higher agrarian land-holding, both none and some land-holding are associated with being uneducated. Relative to the advantaged caste, the two disadvantaged castes are associated with being uneducated, with the risk substantially greater for the Muslim caste. Relative to living near

**Table 6  Hypothesis 4: Broader socio-economic factors associated with women being uneducated.**

| | Model 1: Natal household asset score $n = 2,432^{1}$ $R^2 = 0.210$ | | Model 2: Broader socio-economic factors $n = 2,432^{1}$ $R^2 = 0.363$ | |
|---|---|---|---|---|
| | OR (95% CI) | *p*-value | aOR (95% CI) | *p*-value |
| Women's age (y) | 1.16 [1.12–1.20] | <0.001 | 1.18 [1.14–1.23] | <0.001 |
| Asset score | 1.00 | | 1.00 | |
| Poorest | 11.84 [8.36–16.77] | <0.001 | 8.79 [5.97–12.95] | <0.001 |
| 2nd poorest | 4.01 [2.97–5.40] | <0.001 | 3.60 [2.59–5.01] | <0.001 |
| Mid | 3.18 [2.39–4.23] | <0.001 | 3.16 [2.31–4.34] | <0.001 |
| 2nd richest | 1.68 [1.27–2.22] | <0.001 | 1.68 [1.23–2.28] | <0.001 |
| Richest (ref) | | | 1.00 | |
| Agrarian land | | | | |
| None | | | 2.99 [2.15–4.18] | <0.001 |
| 0.01 to 0.5 hectares | | | 1.79 [1.33–2.42] | <0.001 |
| 0.51 to 0.99 hectares | | | 1.17 [0.84–1.63] | 0.351 |
| ≥1 hectare (ref) | | | 1.00 | |
| Access to big bazaar | | | | |
| <30 min (ref) | | | 1.00 | |
| 30–59 min | | | 1.18 [0.90–1.55] | 0.220 |
| 60–89 min | | | 1.16 [0.83–1.60] | 0.387 |
| ≥90 min | | | 1.62 [1.05–2.51] | 0.031 |
| Caste | | | | |
| Disadvantaged: Dalit | | | 1.66 [1.18–2.35] | 0.004 |
| Disadvantaged: Muslim | | | 7.12 [4.83–10.49] | <0.001 |
| Middle: Janjati, Terai castes | | | 0.99 [0.76–1.29] | 0.927 |
| Advantaged: Yadav, Brahmin (ref) | | | 1.00 | |
| Intercept | 0.04 [0.02–0.09] | <0.001 | 0.01 [0.00–0.02] | <0.001 |

Notes:
  Models include fixed and random effects estimates for geographic clusters and control for trial arm. Grey shading indicates statistically significant associations. aOR, Adjusted Odds Ratio. CI, 95% Confidence Interval.
  [1] $n = 822$ educated (≥1 year schooling) *vs* $n = 1,610$ uneducated.

a big bazaar, living 30–59 min away, but not further distances, is marginally associated with being uneducated.

Model 1 explains 21.0% of the variance in women's education, which is lower than Model 2, which explains 36.3% of the variance. These results support our fourth hypothesis, that poverty, independent of broader socio-economic factors, is associated with women being uneducated.

## Supplementary analysis

A potential reason why poverty may not be associated with women's early marriage, irrespective of the age used to define early marriage, could be that the individual assets owned by households do not actually differ between the wealth quintiles. Table S2 shows, however, that there is substantial variability across the eight individual assets used to produce our composite asset score using PCA. We also include land-holding in this

analysis because it is another marker of wealth in our primarily agrarian population. These individual assets and land ownership matter for daily life and indicate the household's purchasing power. Our results show that asset ownership does indeed differ by wealth levels, nevertheless our analyses described above show that wealth in itself is not associated with women's early marriage in this population.

Given the selection bias in our study of recruiting young, already married and pregnant women, we have restricted the analyses described above to women aged ≥18 years only. With the exception of the association of age, our results are nonetheless similar if we include the full sample of women aged 12–39 years (*n* = 3,379) (Tables S3–S7). The significance of the age variable in these models is a product of the selection effects (as discussed above), and not a cohort effect.

## DISCUSSION

Our study aims to contribute robust evidence on the association between natal household wealth and early marriage. Our results show that if we measure assets in the natal household, we do not support the conventional hypothesis that household poverty is associated with women's early marriage. In our population, relative to the richest households, all other asset level households have an elevated risk of marrying early, and the poorest households do not stand out. Additional analysis shows substantial variability in individual assets and land-holding by asset quintiles, but that wealth in itself is not directly associated with women's early marriage. Our results are somewhat similar to the only other study from Nepal which measured natal household wealth and also did not find the poorest households married daughters the earliest; rather, daughters from the second poorest households were more likely to marry early (*Bajracharya & Amin, 2012*). In contrast, studies from India have either found that poorer households are most likely to marry their daughters early (*Singh & Espinoza Revollo, 2016*), or that wealth of the natal household is not associated with early marriage (*Marphatia et al., 2021b*). Overall, these inconsistent results suggest that we still have a poor understanding of how wealth may shape the timing of women's marriage.

We also found that women's education displaces wealth as a predictor of early marriage, whatever age is used to define early marriage. Relative to secondary-schooled women, women in all other education categories have a higher risk of marrying early. However, there is also a clear gradient, with uneducated women showing the greatest likelihood of early marriage. The association of high wealth with reduced risk of early marriage therefore works through education, and more generally it is women's education level that is directly related to their marriage age. This finding supports the well-established association between women's lower education and their early marriage (*Raj et al., 2014*; *Delprato et al., 2015*; *Sekine & Hodgkin, 2017*; *Marphatia et al., 2020*; *Scott et al., 2021*).

Nonetheless, we found that even in combination, household wealth and women's educational attainment still explain a low proportion of the variance in the likelihood of early marriage models, suggesting that other factors largely drive marriage decisions in this population. Among uneducated women, neither poverty nor broader markers of household disadvantage are associated with early marriage, however 'early' is defined.

Since there was no association of household wealth with early marriage among uneducated women, we suggest that socio-cultural norms that are unconnected with education may be the primary driver of early marriage. For example, if girls are not in school, their primary role in society may be that of a daughter, wife and mother. However, the differences in the age of marriage within uneducated women may also relate to other factors unmeasured by our study.

Where poverty appears to really matter is for the level of education achieved by women, potentially because of the costs associated with schooling, as reported in other studies (*Verma, Sinha & Khanna, 2013*; *Chaudhuri, 2015*; *Samuels et al., 2017*). We found that independent of poverty, both landless and lower agrarian land-holding families are less likely to send their daughters to school. The lack of land may reflect chronic food insecurity, whereas households with some land may prefer daughters to contribute to family income through working on the farm, rather than attending school. Whereas landlessness and the related food insecurity have been associated with both lower schooling and early marriage of girls (*Moock & Leslie, 1986*; *UNICEF, 2014*), our study found that it predicted less schooling, but not early marriage. However, as with wealth, the association of lower agrarian land-holding and early marriage may work through education, in that if girls are not in school, marrying them early may decrease household food and financial burdens (*Maharjan et al., 2012*; *Human Rights Watch, 2016*; *Samuels et al., 2017*).

Like other studies, we also find that disadvantaged castes, especially of the Muslim faith, tend to have lower education (*Stash & Hannum, 2001*; *Sah, 2018*; *Devkota, Eklund & Wagle, 2020*), but this factor was not associated with early marriage. Caste affiliation may be a maker of overall status in society, and therefore act as another marker of access to resources and life opportunities. We find no evidence of the natal household's proximity to big bazaars mattering for early marriage, and there is also no clear relationship between distance from bazaar and lack of education.

## Implications

Our results have implications for research, policy and practice. First, there is an urgent need for informative objective data measuring wealth in women's natal households, ideally before they marry, and also in women's marital households, ideally at the time of their marriage. This approach would provide much needed evidence on whether natal household poverty is indeed associated with the likelihood of early marriage and whether women marry into households of similar wealth levels.

These data-related issues, and in particular the inadequate understanding of the association of wealth and marriage age, may partly explain the inconsistent results of interventions targeting poverty as the main driver of early marriage. A recent systematic review found conditional cash transfers (CCTs) to keep girls in school were more effective in delaying marriage than those directly targeting delayed marriage or poverty (*Malhotra & Elnakib, 2021*). In Bangladesh, CCTs supported girls to stay in school for longer, but delayed marriage for only the youngest girls (aged 12–14 years) living in the poorest district (*Amin, 2007*). In India, another intervention gave CCTs to parents if their

daughters had reached the age of 18 years unmarried. Compared to a control group, the girls who participated in this intervention stayed in school up to grade eight, and were in fact more likely to marry just after the 18 year cut-off stipulated by the intervention (*Nanda et al., 2015*). In both interventions, the cash transfers was primarily used by parents to pay the higher cost of dowries demanded by marital households for older, more educated girls (*Amin, 2007*; *Nanda et al., 2015*).

Second, we need to better understand the different factors increasing the risk of marrying at different ages, for women of different education levels. Reducing the costs of schooling (*e.g.* fees, learning materials, offering scholarships) and improving the quality of education may help girls to stay in school for longer and also delay marriage, thereby achieving several of the Sustainable Development Goals (*Muchomba, 2021*). Further research is needed to understand whether the expansion of free state provision of secondary education would have a sustained impact on delaying marriage. However, any school-based efforts will miss the girls who never went to school in the first place or have already dropped out and married.

Third, there needs to be a collective shift in societal gendered norms and the value attributed to girls and women in society (*Maertens, 2013*; *Bicchieri, Jiang & Lindemans, 2014*; *Marphatia, Amable & Reid, 2017*). Changing norms is difficult and slow, as shown by social interventions that did not succeed in delaying marriage age in India (*Prakash et al., 2019*; *Ramanaik et al., 2020*). Moreover, in populations where women marry very early, it may initially be more realistic to delay marriage by 1 year at a time. In our study, this would mean first delaying marriage from 15 to 16 years, and then from 16 to 17 years. 'Nudging' populations towards a slightly later marriage age for women may therefore lead to more substantial secular changes over time. Whilst this may sit uncomfortably with the human rights constituencies advocating for the 18-year minimum marriage age, ignoring these practiced norms may render us even further from the common goal of delaying marriage overall (*Schaffnit, Urassa & Lawson, 2019*). Delaying marriage, even by 1 year, will inevitably delay the age at first child bearing. Early marriage must therefore be seen as a critical concern for public health (*Marphatia, Amable & Reid, 2017*).

### Limitations

Our study has some limitations. With cross-sectional data, we can only investigate associations and not causality. Although we include education in early marriage models, like other studies, we do not know the direction of this association. We did not have data on the amount of dowry paid by women's natal household, nor whether it accelerated the timing of their marriage. Our study involved only married pregnant women and we found that women measured in their natal homes during pregnancy differed from those who remain in their marital homes. However, these differences were small. As we do not have matching data on pregnant women in their marital homes we cannot explore the other ways in which they may differ. In our population, women are also very unlikely to continue their education after marriage, and the vast majority of married women are likely to have children, and it is common in Nepal get pregnant in the first 2 years after

marriage. Our asset score was measured in the natal household after, and not at, marriage. To address this, our asset score excluded items which could have been purchased after marriage. Despite these limitations, our study benefited from a large sample size and the unique data on several different socio-economic variables as well as women's education. The associations between poverty, education and marriage age, and poverty and lack of education identified in our study are likely to be widely applicable, especially to similar Madhesi populations living around the bordering regions of India and Nepal.

## CONCLUSION

Our study is unique in having objective data on assets and broader markers of disadvantage measured in women's natal household. These data enable us to conduct robust and appropriate investigations of the association of natal household wealth with early marriage, using different age thresholds to define 'early'. We also investigate the association of education with the likelihood of early marriage, and whether poverty is associated with early marriage in uneducated women who comprise two-thirds of our sample. Finally, we investigate whether poverty is associated with the likelihood of women having no formal education. Whilst we do not find that natal household poverty predicted early marriage in rural lowland Nepal, further research is required in other populations to establish whether this association is apparent more generally.

## ACKNOWLEDGEMENTS

We thank the women and their families for participating in the trial, and for allowing us to measure their newborns. The Public Health Offices of Dhanusha and Mahottari Districts in Nepal supported implementation of the trial. We also thank Mother and Infant Research Activities (MIRA, Nepal) staff for data collection, and the UCL Institute for Global Health team for their support (see *Saville et al., 2018* for details).

### Funding

This work was supported by the Leverhulme Trust (Grant Number: RPG-2017-264) and National Institute for Health Research (NIHR) Great Ormond Street Hospital Biomedical Research Centre. Funding for the LBWSAT was provided by: Department for International Development (DFID) South Asian Research Hub (Grant Number: PO 5675). The funders had no role in study design, data collection and analysis, decision to publish, or preparation of the manuscript.

### Grant Disclosures

The following grant information was disclosed by the authors:
Leverhulme Trust: RPG-2017-264.
National Institute for Health Research (NIHR) Great Ormond Street Hospital Biomedical Research Centre.
Department for International Development (DFID) South Asian Research Hub: PO 5675.

## Competing Interests

The authors declare that they have no competing interests.

## Author Contributions

- Akanksha A. Marphatia conceived and designed the experiments, analyzed the data, prepared figures and/or tables, authored or reviewed drafts of the paper, and approved the final draft.
- Naomi M. Saville conceived and designed the experiments, performed the experiments, analyzed the data, authored or reviewed drafts of the paper, and approved the final draft.
- Dharma S. Manandhar performed the experiments, authored or reviewed drafts of the paper, and approved the final draft.
- Mario Cortina-Borja conceived and designed the experiments, analyzed the data, prepared figures and/or tables, authored or reviewed drafts of the paper, and approved the final draft.
- Jonathan C. K. Wells conceived and designed the experiments, analyzed the data, authored or reviewed drafts of the paper, joint senior author, and approved the final draft.
- Alice M. Reid conceived and designed the experiments, authored or reviewed drafts of the paper, joint senior author, and approved the final draft.

## Human Ethics

The following information was supplied relating to ethical approvals (*i.e.*, approving body and any reference numbers):

Research ethics approval to conduct the Low Birth Weight South Asia Trial (LBWSAT) was granted by the Nepal Health Research Council (108/2012) and University College London (UCL) Research Ethics Committee (4198/001). Further ethical approval for secondary analyses of LBWSAT data for this analysis was granted from the Nepal Health Research Council (292/2018), the Research Ethics Committees at UCL (0326/015) and the University of Cambridge (1016).

## Field Study Permissions

The following information was supplied relating to field study approvals (*i.e.*, approving body and any reference numbers):

Research ethics approval to conduct the Low Birth Weight South Asia Trial (LBWSAT) was granted by the Nepal Health Research Council (108/2012) and University College London (UCL) Research Ethics Committee (4198/001). Village Development Committee secretaries consented for villages to participate in the trial. Women gave written consent and guardians consented for villages to participate in the trial.

## Data Availability

The raw data used in the analyses are available in the Supplemental File.

## Supplemental Information

Supplemental information for this article can be found online at http://dx.doi.org/10.7717/peerj.12324#supplemental-information.

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
