# Peer review of "Quantifying the association of natal household wealth with women’s early marriage in Nepal"

_PeerJ, doi:10.7717/peerj.12324_

## Round 0.1 · original submission · Minor Revisions

The manuscript was reviewed by three experts in the field and their report is detailed below

Please address the comments of the reviewers prior to any further consideration of your manuscript.

·

Basic reporting

In the introduction (line 87), the term “natal” or “natal household” should be defined first and if possible clarified by providing more general term with similar meaning to ease readers in understanding the whole meaning of the sentences the author wrote.

Experimental design

In the Statistical Method section (line 327), it would be better to present the equation of logistic regression model with random effect stated in line 340 and explain the main difference between this model with the ordinary one.

Validity of the findings

No comment

Additional comments

I hope I can read this paper in PeerJ soon.

·

Basic reporting

I have read the paper with great interest and found it interesting. Early marriage in this part of the world needs to be explored more and the authors did an excellent job overall. However, the paper requires some language editing. The authors should get the manuscript revised by someone who excels in English language for this purpose.

Experimental design

The quantitative experimental design is well organized and hypotheses were tested properly.

Validity of the findings

The findings ensure valid results, but the discussion section needs to be revised. It lacks sound references to compare and contrast implications for the findings.

Additional comments

Overall, it is a good attempt, but the authors need to ensure the language edits and a revision for the discussion section.

Reviewer 3 ·

Basic reporting

This article is well written and structured. The tables are sufficient and the authors note availability of raw data as a supplement. There are a few minor issues that need to be clarified as the meaning is not clear:

On line 109 the authors state: “Attributing wealth to the correct household is crucial in this context, because it reflects not only the family’s socio-economic status but also its spatial niche”

I do not understand what the authors mean here by spatial niche.

I am surprised that there was little mention of dowry in the background considering how considerable dowry can be and presumably is connected to natal family wealth. Perhaps the authors can comment on why dowry is not given much attention in the background.

Experimental design

My main issue with this paper is the sampling of married women who are pregnant. What do we miss with married women who are not pregnant? Might there be a relationship between pregnancy and education and thus only including pregnant women distorts the relationship between early marriage and education—women who are not pregnant might have continued education or delayed childbearing to achieve education or livelihood goals? It would be helpful if the authors could add more justification for why this sample is appropriate to answer their main question(s) and consider how their results might change if their sample was married women (including those not pregnant).

Authors correctly point out that wealth typically measured in marital household but they do not mention that this may because it is difficult to measure natal wealth when girls are married and move to marital homes and this is where they are ‘found’ in child marriage studies. The authors note on line 380 they excluded “17,335 women who were interviewed in their marital home” --this represents the majority of women recruited into the study. They also note “Questionnaires were administered orally to 25,090 married pregnant 239 women aged 10-49 years in the home that they were residing in during pregnancy (Saville et al., 240 2016).” The supplemental table provided shows that these groups differ on a number of indicators that may be associated with the outcomes of interest. How else may girls/women (or their natal families) differ when those girls/women are able and supported to return to their natal homes during pregnancy?

The authors acknowledge this issue on line 405: “Whilst these results show that women who went to their natal homes during pregnancy were different than those who stayed in their marital homes, they still represent an important sub group of women with rare data on the socio-economic characteristics of their natal homes.”
I think the authors need to justify this more as their results are not generalizable to women who remain in their marital homes during pregnancy, which likely represents the majority of women married as children globally.

The methods are adequately described and the research appears to be well conducted.

Ethics appear to be well thought through and adhered to.

Validity of the findings

The authors need to be careful not to use language that implies causality when their results only support an association. For example, in the conclusion the authors state: “When assets are measured in the natal household in this population, there is no support for the conventional hypothesis that household poverty drives families to marry off their daughters early, but it does determine whether they go to school.” -- this statement implies some causality but this study cannot measure causality.

Similarly, line 541: “Our results show that if we measure assets in the natal household, then we do not support the conventional hypothesis that household poverty directly drives families to marry off their daughters early.”

Additional comments

No comment

---

## Round 0.2 · accepted · Accept

The manuscript has been reviewed by two experts in the field and the authors have satisfactory addressed their comments.